# The Use of Allogeneic Hematopoietic Stem Cell Transplantation in Primary Myelofibrosis

**DOI:** 10.3390/jpm12040571

**Published:** 2022-04-02

**Authors:** Heather R. Wolfe, Mitchell E. Horwitz, Lindsay A. M. Rein

**Affiliations:** 1Department of Medicine, Duke University Medical Center, Durham, NC 27707, USA; 2Division of Hematologic Malignancies and Cellular Therapy, Department of Medicine, Duke University Medical Center, Durham, NC 27707, USA; mitchell.horwitz@duke.edu (M.E.H.); lindsay.magura@duke.edu (L.A.M.R.)

**Keywords:** primary myelofibrosis, myeloproliferative neoplasms, hematopoietic stem cell transplantation, conditioning regimens, donor selection

## Abstract

Primary myelofibrosis (PMF) is a BCR-ABL1 negative myeloproliferative neoplasm characterized by clonal proliferation of myeloid cells. This leads to reactive bone marrow fibrosis, ultimately resulting in progressive marrow failure, hepatosplenomegaly, and extramedullary hematopoiesis. PMF is considered the most aggressive of the BCR-ABL1 negative myeloproliferative neoplasms with the least favorable prognosis. Constitutional symptoms are common, which can impact an individual’s quality of life and leukemic transformation remains an important cause of death in PMF patients. The development of the Janus kinase 2 (JAK2) inhibitors have provided a good option for management of PMF-related symptoms. Unfortunately, these agents have not been shown to improve overall survival or significantly alter the course of disease. Allogenic hematopoietic stem cell transplantation (allo-HSCT) remains the only curative treatment option in PMF. However, allo-HSCT is associated with significant treatment-related morbidity and mortality and has historically been reserved for younger, high-risk patients. This review examines patient, disease, and transplant-specific factors which may impact transplant-related outcomes in PMF. Through the vast improvements in donor selection, conditioning regimens, and post-transplant care, allo-HSCT may provide a safe and effective curative option for a broader range of PMF patients in the future.

## 1. Introduction

Primary myelofibrosis (PMF) is a BCR-ABL1 negative myeloproliferative neoplasm characterized by clonal proliferation of myeloid cells. This leads to reactive bone marrow fibrosis resulting in progressive marrow failure, hepatosplenomegaly, and extramedullary hematopoiesis. The median age at the time of diagnosis is 67 years; however, PMF affects both middle-aged and older adults [1]. The clinical presentation of PMF can be heterogenous, as up to 30% of patients are asymptomatic at the time of diagnosis. However, patients can present with constitutional symptoms or symptoms related to underlying anemia or splenomegaly. Laboratory abnormalities are a hallmark of this condition but can vary depending on factors such as degree of marrow fibrosis, splenomegaly, and stage of disease. Anemia is common and occurs in greater than 50% of patients while white blood cell and platelet counts can be variable [2,3]. Secondary myelofibrosis, or myelofibrosis which arises from polycythemia vera (PV) or essential thrombocythemia (ET), presents unique challenges. Patients are considered particularly high-risk due to the increased risk of leukemic transformation and inferior overall survival (OS) rates compared to primary myelofibrosis. 

The initial diagnosis of PMF requires a bone marrow biopsy and both cytogenetic and molecular analysis. The 2016 World Health Organization (WHO) highlights the major and minor diagnostic criteria for PMF. Major criteria include: (1) megakaryocytic proliferation and atypia, (2) not meeting other WHO criteria for other myeloid neoplasms, and (3) the presence of JAK2, CALR, or MPL mutations, or in the absence, the presence of another clonal marker. In addition, at least one minor criterion is required, including anemia, leukocytosis, palpable splenomegaly, elevated lactate dehydrogenase, or leukoerythroblastosis [4,5]. BCR-ABL1 testing is required to rule out chronic myeloid leukemia (CML). Often, the bone marrow can be difficult to aspirate in PMF, leading to a “dry” tap. Some degrees of fibrosis on marrow evaluation is seen in almost all patients with PMF and grading is important to distinguish between pre-fibrotic myelofibrosis and PMF as prognosis and management often differ. Some patients may have a hypercellular marrow with limited fibrosis consistent with the cellular phase of PMF. As with other myeloproliferative neoplasms, PMF has been associated with mutations in JAK2, CALR, and MPL in approximately 60%, 20%, and 5% of patients, respectively [6,7,8,9]. Approximately 10% of patients with PMF are considered “triple-negative” and have no identifiable mutation [10].

The complications of PMF include constitutional symptoms that may impact quality of life, bone marrow failure, symptomatic hepatomegaly or splenomegaly, and leukemic transformation. While leukemic transformation is rare, occurring in approximately 4% of individuals, it is an important cause of death in PMF patients [11]. PMF is considered the most aggressive of the BCR-ABL1 negative myeloproliferative neoplasms with the least favorable prognosis. Notably, the clinical course is highly variable. Based on multivariable analysis from prior studies; older age, anemia, leukocytosis, thrombocytopenia, circulating blasts, degree of marrow fibrosis, constitutional symptoms, and transfusion dependence have been associated with inferior survival. The differing clinical courses and identifiable clinical risk-factors have led to the development and evolution of several different prognostic systems. Currently, there is no one preferred prognostic tool. The dynamic international prognostic scoring system (DIPSS) is a unique tool in that it can be used any time during the clinical course of disease [12]. Other models such as the genetically inspired prognostic scoring system (GIPSS) and mutation-enhanced international prognostic score system (MIPSS70+) have incorporated cytogenetic and molecular data, which may improve the prognostic value. The MIPSS70+ was specifically developed for patients ≤ 70 years of age to identify high-risk patients who would benefit from transplantation [13]. All prognostic tools can identify potentially high-risk patients for which treatment may differ from the low-risk individuals. The established PMF prognostic scoring tools are compared in Table 1. Outcomes differ dramatically between the risk groups with a median OS of >10 years in low-risk patients, 4–7 years in intermediate risk, and 2–4 years in the high-risk group [12,13,14,15]. Mutation status has also been found to impact outcomes with patients with CALR mutation (median OS 17.7 years) having the most favorable outcomes when compared to JAK2-mutant (median OS 9.2 years) or triple-negative patients (median OS 3.2 years) [9]. The myelofibrosis secondary to PV and ET prognostic model (MYSEC-PM) was designed for individuals with secondary myelofibrosis [16]. When compared to the DIPSS, the MYSEC-PM may enhance risk stratification and allow for better counseling for these high-risk patients with respect to post-transplant outcomes [17].

Treatment of PMF is based upon an individual’s risk assessment (Table 2). The National Comprehensive Cancer Network (NCCN) guidelines define low-risk disease as DIPSS ≤ 2, DIPSS-Plus ≤ 1, or MIPSS-70+ version 2.0 ≤ 3 [18]. Asymptomatic, low-risk disease can be typically managed with observation alone or supportive care as more aggressive interventions and therapeutics have not been shown to alter disease course. Symptom-directed therapies, including the Janus kinase 1/2 (JAK2) inhibitors, peginterferon alfa-2a, or hydroxyurea should be considered. The JAK2 inhibitor, ruxolitinib, is efficacious in reducing symptoms and splenomegaly in patients with PMF. Its efficacy has been shown independent of JAK2 mutation status. Hematological adverse events, such as anemia and thrombocytopenia, are common with initiation of the JAK2 inhibitors. However, studies have shown that a decline in hemoglobin values from baseline, while on ruxolitinib therapy, does not impact OS. Transient treatment-related anemia should not lead to premature drug interruption, discontinuation, or dose titration during the first few months of therapy [19,20].

Despite improvement in symptoms, ruxolitinib has not been shown to prolong survival in PMF [21]. Fedratinib is an oral, potent JAK2 inhibitor which has been approved for treatment of intermediate-2 or high-risk PMF [25]. Hydroxyurea is a treatment option for patients who are ineligible for transplantation and are not candidates for ruxolitinib or fedratinib. Hydroxurea can improve splenomegaly, thrombocytosis, leukocytosis, bone pain, and constitutional symptoms in patients with PMF. However due to drug-associated cytopenias, hydroxurea is not beneficial in cytopenic myelofibrosis [22].

High-risk PMF, typically classified as intermediate-2 or high-risk, is defined as DIPSS > 2, DIPSS-Plus > 1, or MIPSS-70+ version 2.0 > 3 [12,13,14]. Patients with high-risk disease should be considered for early allogeneic bone marrow transplantation rather than symptom-directed therapy due to their poor OS and increased risk for leukemic transformation. High-risk patients who are transplant ineligible should be considered for clinical trial or symptom-directed therapies with the JAK2 inhibitors. In addition, patients with cytopenic myelofibrosis should be considered for clinical trial or early allogenic bone marrow transplantation as the cytopenias often preclude use of other therapies. The treatment discussion for patients with intermediate-risk disease is more complicated due to the variable clinical courses and risk for transplant-related morbidity and mortality.

## 2. Use of Bone Marrow Transplantation in PMF

Allogenic hematopoietic stem cell transplantation (allo-HSCT) is the only curative treatment option in PMF. However, allo-HSCT is associated with significant treatment-related morbidity and mortality due to toxicity from conditioning regimens, graft failure, and potential development of graft versus host disease (GVHD). The use of conventional conditioning strategies in PMF has been associated with significant mortality, with conditioning-related mortality rates of 10–20% [26,27,28]. In addition, graft failure and relapse remain significant issues post-transplantation [29,30].

Historically, allogenic transplants have been reserved for younger, higher-risk patients, with a suitable donor for which the benefit of transplant likely outweighs the potential risk. As noted above, the NCCN guidelines recommend upfront consideration for allo-HSCT for patients with higher-risk disease defined as DIPSS > 2, DIPSS-Plus > 1, or MIPSS-70+ version 2.0 > 3 who have an estimated median survival of 2–4 years [18]. The support for this recommendation comes from a retrospective multicenter study of patients with PMF in the pre-JAK inhibitor era, which compared outcomes in patients who underwent allo-HSCT versus those who received conventional therapy. Patients with intermediate-2 or high-risk DIPSS scores clearly benefited from transplant; however, low-risk patients did not derive benefit [31].

Over the last few decades, there have been significant improvements in patient and donor selection, conditioning regimens, and post-transplant supportive care. More recent data from an experienced transplant center demonstrated a relapse incidence of 11%, 5-year non-relapse mortality (NRM) of 34% and 7-year survival of 61% in related and unrelated donor transplants in patients with myeloproliferative neoplasms [28]. This has led to a paradigm shift with experts now considering allo-HSCT as a curative therapy in younger, fit patients in all risk categories as opposed to just higher risk patients. Bacigalupo et al. demonstrated that the five-year OS was improved in low-risk patients versus high-risk patients following allo-HSCT due to a higher transplant-related mortality (TRM) and higher relapse-related death in high-risk individuals [32]. This has been redemonstrated in an additional study which showed improvements in rates of NRM, disease relapse, and overall mortality with allo-HSCT in low and intermediate-1 disease [33]. Pursuing transplant earlier in the disease course can leverage a patient’s younger age, favorable performance status, and less exposure to prior therapies with the goal to improve post-transplant complications. We will further examine patient, disease, and transplant-specific factors which may impact transplant-related outcomes.

### 2.1. Patient-Related Factors

Age has been shown to be the most significant factor that impacts allo-HSCT outcomes in myeloproliferative neoplasms [30,34]. The inverse correlation between age and outcome is likely related to comorbid conditions and other age-related metabolic disorders that can leave patients vulnerable to treatment-related toxicity [28,35]. Samuelson et al. studied a highly selected group of older patients (60–78 years of age) with primary or secondary myelofibrosis who underwent allo-HSCT. The group demonstrated favorable time to engraftment, rates of GVHD, progression free survival (PFS), and OS, comparable to what has been previously reported in younger patients [36]. This study suggests that advanced chronologic age may not be an absolute contraindication to allo-SCT but focusing on well-selected older adults with good performance status and minimal comorbidities may provide favorable outcomes.

During the allo-SCT evaluation, patients will undergo a comprehensive history and physical to evaluate for comorbid conditions, overall physical fitness, and necessary social support. Estimated NRM and OS can be assessed using established the validated Hematopoietic Cell Transplantation-specific comorbidity index (HCT-CI) [37]. This can be helpful to aid in the risk and benefit discussion with the patient; however it does not take into consideration the potential transplant-related morbidity which may be similarly devastating to patients.

### 2.2. Disease-Related Factors

Splenomegaly is a common finding in patients with PMF and results from extramedullary hematopoiesis most commonly in the spleen. Notably, this can also occur in the liver and other organs. In addition to causing symptoms and affecting quality of life, splenomegaly has been associated with graft failure and delayed hematologic recovery following allo-SCT [35,38,39]. Historically, pre-transplant splenectomy has been studied but can be associated with significant peri-operative complications and mortality rates of 5–10% [40]. Alternatively, pre-transplant splenic irradiation has been used; however, it is reserved for patients with symptomatic splenomegaly despite the use of JAK inhibitors. Data is limited with regards to the utility of splenic irradiation in this population. With the discovery of the JAK2 inhibitors, the use of ruxolitinib has been associated with significant improvements in disease-associated symptoms and splenomegaly [21,23]. Ruxolitnib is frequently used in the pre-transplant setting to reduce disease-symptoms to improve performance status and symptomatic splenomegaly to improve engraftment [41,42]. Studies using JAK inhibitors prior to allo-HSCT have demonstrated both safety, similar disease-free survival (DFS) and OS, and a trend towards a lower rate of disease relapse compared to those without prior JAK inhibitor exposure [42,43].

Molecular profiling has also led to the identification of other high-risk individuals who may or may not benefit from early transplantation. It is established that patients with CALR type-1/like mutations have improved survival compared to patients with CALR type-2/like, MPL, or JAK2 mutations [44]. As noted above, approximately 10% of patients with PMF have no identifiable mutation and are considered “triple negative”. Triple negative patients have inferior outcomes, with a median OS in one study of 3.2 years [9]. Additionally, high-risk non driver mutations, such as the presence of EZH2, ASXL1, IDH1/2, and SRSF2, are associated with inferior OS and DFS [45]. Some suggest that early transplantation in individuals with non-driver mutations should be considered in younger patients with good donors as their outcomes remain poor [46].

### 2.3. Conditioning Selection

Conventional myeloablative regimens are associated with significant toxicity and high treatment-related mortality rate with 1-year NRM ranging from 20–48% [26,47]. A large study examined 286 patients with PMF who underwent allo-HSCT. Most patients received myeloablative conditioning (MAC) with either total body irradiation (TBI) and cyclophosphamide or busulfan and cyclophosphamide. The 100-day transplant-related mortality (TRM) was 20–30% and the 1-year OS was approximately 50–60% [47]. Conditioning regimens using fludarabine plus two alkylating agents, such as thiotepa and busulfan, have been studied in patients with myelofibrosis and been shown to reduce the risk of relapse; however, they are associated with significant toxicity [48].

A movement towards reduced-intensity conditioning (RIC) has led to reductions in therapy-related morbidity and mortality and has provided a transplant option in older and less fit individuals. Many studies have shown successful engraftment and durable remissions in patients > 60 years of age with RIC strategies in patients with myelofibrosis [29,36,47]. More recently, Robin et al. compared two commonly used RIC regimens in myelofibrosis: fludarabine-busulfan (FB) and fludarabine-melphalan (FM) in a retrospective study. The 7-year OS was >50% in both groups. The FM regimen appeared to be more toxic with higher rates of acute GVHD. Multivariate analyses demonstrated a lower relapse rate in patients who received FM (hazard ratio, 9.21; *p* = 0.008); however, a trend towards lower NRM in patients who received FB (hazard ratio, 0.51; *p* = 0.68) was seen. They found no significant differences between the conditioning regimens with regards to progression free or OS [49]. Accordingly, RIC may be a reasonable option for older (>60 years of age) or less fit patients to maximize benefit while minimizing potential risk.

### 2.4. Donor Selection

During the initial transplant evaluation, potential related-donors can be identified and typed and if needed, an unrelated-donor search can be initiated. Due to concerns for risk of graft rejection and graft-versus-host disease in patients with PMF, traditionally related donors were preferred. In the large study of PMF patients who underwent allo-HSCT described above, 56% received matched sibling donor transplants, 9% received matched other related donor transplants, and 35% received unrelated donor (URD) transplantations. At 100 days, transplant-related mortality was similar between the matched sibling and matched other transplant groups; however, unrelated donor transplants had higher TRM and higher rates of graft failure compared to matched related donor (MRD) transplantations. Relapse rates were similar between the groups. They found no significant difference in acute or chronic GVHD, DFS, or OS at five years [47].

However, a subset of transplant eligible patients do not have a suitably matched donor leading to the investigation of alternative sources. In 2019, the European Society of Bone Marrow Transplantation published a study evaluating the safety and efficacy of mismatched related donors, also referred to as haploidentical donors, in 56 myelofibrosis patients. This study demonstrated favorable rates of engraftment, acceptable rates of GVHD, and a 2-year OS rate of 56% [50]. When compared with matched donors, haploidentical donors had a higher NRM at 1 year (12% versus 38%, respectively) [29]. The use of umbilical cord blood has also been studied in Japan for high-risk myelofibrosis patients with acceptable rates of engraftment, GvHD, and a 2-year OS of 44% [51]. However, the data continues to support the use of matched related donors when available.

### 2.5. Use of JAK Inhibitors

JAK inhibitors have been studied in the pre-transplant setting in hopes of reducing tumor burden and spleen size while simultaneously decreasing the degree of constitutional symptoms [43]. A recent large retrospective study demonstrated a trend towards lower graft failure rate and relapse rate, and improved event-free survival in patients who had received ruxolitinib prior to allo-HSCT. These trends were significant in patients with ongoing spleen response to ruxolitinib going into transplantation. These favorable effects related to ruxolitinib are attributed to the decrease in spleen size, leading to improved engraftment and graft function, as well as improvement in pre- and peri-transplant constitutional symptoms [52]. A study published in 2018 demonstrated that continuing a low dose of ruxolitinib at 5 mg twice a day until engraftment potentially reduced the risk of graft failure and incidence of GVHD within the first 100 days of transplant [53].

Reactivation of latent infection remains an important consideration in patients receiving ruxolitinib. Patients who undergo allo-HSCT are at particularly high risk of cytomegalovirus (CMV) and Epstein-Barr virus (EBV) reactivation with the use of ruxolitinib. In the above study, CMV reactivation occurred in 41% in patients receiving low-dose ruxolitninb, a significantly higher rate than what is reported in non-transplant patients receiving ruxolitinib [21,53]. To balance the potential benefits with the potential infectious complications, ruxolitinib is often discontinued prior to conditioning. A taper is usually required to prevent a cytokine rebound and symptoms associated with drug discontinuation [42]. Following transplant, the optimal use of ruxolitinib remains unknown. Ruxolitinib has been studied and is now being clinically used for steroid refractory acute GVHD [54,55]. Of note, ruxolitinib is the only JAK inhibitor which has been studied in the pre-transplant and post-transplant setting. Further studies evaluating the efficacy and safety of the selective JAK2 inhibitor, fedratinib, are needed as fedratinib may have lower rates of infectious complications.

## 3. Recommendations

Allo-HSCT remains an important curative option for patients with PMF. When assessing a PMF patient for transplantation, focus should be placed on: (1) pre-transplant symptom burden and quality of life, (2) age, (3) comorbidities, (4) disease-specific factors, (5) functional status, and (6) availability of related donors.

The use of established prognostic tools, such as the DIPSS-Plus or MIPSS-70+ version 2.0, which utilize cytogenetic and mutational data, should be used to risk stratify patients who may derive the greatest benefit from allo-HSCT. Transplantation should always be considered for individuals with intermediate-2 and high-risk disease. Patients with intermediate-1 risk disease in the presence of other poor prognostic factors, such as high-risk cytogenetics or mutations, should also be considered for transplantation. Curative allo-HSCT should be discussed and considered in young patients, regardless of their risk score, to leverage age, favorable performance status, and less exposure to prior therapies. Of note, the current established prognostic tools have been studied in cohorts of patients in the pre-ruxolitinib era, which can lead to uncertainty regarding true survival and outcomes. A recent prognostic scoring system has been proposed to better select patients for transplantation. The myelofibrosis transplant scoring system (MTSS) uses a variety of patient, disease, and transplant-specific factors to help predict post-transplant outcomes. In addition to age, constitutional symptoms, and mutation status which is often accounted for in the established prognostic tools described above, they add transplant-specific factors such as performance status, donor HLA-match, CMV status, and use of ruxolitinib before transplant to further improve prognostic ability [56]. While encouraging, further studies are needed to establish its large-scale applicability.

Optimal donor selection remains an important factor with trends towards improved outcomes with MRD and MUD transplants. However, there is increasing evidence supporting the use of alternative donors such as umbilical cord blood and haploidentical donor cells. Alternative donors provide an important option for those who would have been historically ineligible due to donor availability.

In young, fit patients, myeloablative regimens such as busulfan and cyclophosphamide are recommended [27,28]. However, in older patients or those with significant comorbid conditions, reduced-intensity conditioning with fludarabine and busulfan or fludarabine melphalan should be considered [49,57,58]. We recommend that JAK inhibitors be continued but tapered through a 10-to-14-day period either before or upon completion of the conditioning regimen to gain the benefits of JAK inhibition while reducing the risk of infection, including CMV reactivation. Further studies are needed to evaluate the use of JAK inhibitors post-transplant to reduce the risk of relapse.

Unfortunately, despite the dramatic improvements in transplantation over the years, many patients with PMF who undergo transplant evaluation are considered ineligible due to age, comorbidities, or other factors and should be considered for clinical trial or symptom-directed therapies.

## 4. Conclusions

Despite the significant risk of morbidity and mortality associated with allo-HSCT, it remains the only curative option for patients with PMF. Historically, transplantation was reserved for young patients with high-risk diseases. However, improvements in conditioning regimens, supportive care, and improvements in alternative donor transplantation has led to expanding the eligibility to include older patients and those without a fully matched donor. The incorporation of the JAK inhibitors has also provided improvements in PMF-related symptoms and pre-transplant performance status, potentially leading to improved transplant outcomes. Future studies are needed to study the potential benefit of early versus late transplantation in lower-risk PMF patients as well as optimal use of JAK inhibitors pre-, peri-, and potentially post-transplantation.

## Figures and Tables

**Table 1 jpm-12-00571-t001:** Established Prognostic Models in Primary Myelofibrosis.

	Dynamic International Prognostic Scoring System (DIPSS) [12]	Dynamic International Prognostic Scoring System (DIPSS-Plus) [14]	Genetically Inspired Prognostic Scoring System (GIPSS) [15]	Mutation-Enhanced International Prognostic Scoring System Plus Karyotype (MIPSS70+ Version 2) [13]
Age	>65 years = 1			
Constitutional Symptoms	Yes = 1 No = 0			Yes = 2 No = 0
Anemia	Hgb < 10 g/dL = 2	Red cell transfusion need = 1		Hgb < 8 g/dL (women), <9 g/dL (men) = 2 Hgb 8–9.9 g/dL (women), 9–10.9 g/dL (men) = 1 Hgb > 10 (women), Hgb > 11 (men) = 0
White Blood Cell Count (WBC)	WBC ≥ 25,000/µL = 1			
Platelet Count		<100,000/µL = 1		
Circulating Blasts	≥1% = 1			2% = 1 <2% = 0
Karyotype		Unfavorable karyotype * = 1	Very-high risk ^∆^ = 2 Unfavorable ^∆^ = 1 Favorable ^∆^ = 0	Very-high risk ^∆^ = 4 Unfavorable ^∆^ = 3 Other ^∆^ = 0
Presence of Driver Mutations			Absence of type 1-like CALR = 1 ASXL1 = 1 SRSF2 = 1 U2AF1 Q157 = 1	Absence of type 1-like CALR= 2 >2 High Molecular Risk (HMR) Mutations ^‡^ = 3 1 HMR mutations ^‡ ^= 2 0 HMR mutations ^‡ ^= 0
Interpretation (Median overall survival)	Low risk: 0 (not reached) Intermediate-1: 1–2 points (14.2 years) Intermediate-2: 3–4 points (4 years) High risk: 5–6 points (1.5 years)	Low risk: 0 (185 months) Intermediate-1: 1 point (78 months) Intermediate-2: 2 points (35 months) High risk: 3 points (16 months)	Low risk: 0 (26.4 years) Intermediate-1: 1 point (10.3 years) Intermediate-2: 2 points (4.6 years) High risk: ≥3 points (2.6 years)	Very low risk: 0 (not reached) Low risk: 1–2 points (10.3 years) Intermediate risk: 3–4 points (7 years) High risk: 5–8 points (3.5 years) Very high risk: ≥9 points (1.8 years)

* Unfavorable karyotype includes: (1) Complex karyotype rearrangements; (2) One or two abnormalities that include +8, -7/7q-, i(17q), -5/5q-, 12p-, inv(3), or 11q23. ^∆^ Cytogenetics: Very high risk (VHR): Single/multiple abnormalities of -7, i(17q), inv(3)/3q21, 12p-/12p11.2, 11q-/11q23, or other autosomal trisomies not including +8/+9 (e.g., +21, +19). (1) Favorable: Normal karyotype or sole abnormalities of 13q-, +9, 20q-, chromosome 1 translocation/duplication or sex chromosome abnormality including -Y. (2) Unfavorable: All other abnormalities. ^‡^ High molecular risk (HMR) mutations: ASXL1, SRSF2, EZH2, IDH1, IDH2, U2AF1 Q157.

**Table 2 jpm-12-00571-t002:** Initial Treatment for PMF According to Disease Risk.

	Low-Risk Disease (Asymptomatic)	Low-Risk Disease (Symptomatic)	Higher-Risk Disease ^•^
Young and fit patients	Clinical Trial Observation	Clinical Trial Ruxolitinib (no changes in overall survival) [21] Peginterferon alfa-2a Hydroxyurea (40% of patients with reduction in spleen size) [22] Consider early allo-HSCT	Referral for allo-HSCT Clinical Trial JAK Inhibitor Ruxolitinib (42% of patients with reduction in spleen size) [23] Fedratinib (36% of patients with reduction in spleen size) [24]
Older Patients or those with significant comorbid conditions	Clinical Trial Observation	Clinical Trial Ruxolitinib (no changes in overall survival) [21] Peginterferon alfa-2a Hydroxurea (82% of patients with improved constitutional symptoms) [22]	Consider allo-HSCT Clinical Trial JAK Inhibitor Ruxolitinib (46% of patients with improved symptoms) [23] Fedratinib (34% of patients with improved symptoms) [24]

^•^ Defined as DIPSS > 2, DIPSS-Plus > 1, or MIPSS-70+ version 2.0 > 3.

## Data Availability

Not applicable.

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
