# Peer review of "The Use of Allogeneic Hematopoietic Stem Cell Transplantation in Primary Myelofibrosis"

_jpm, 2022, doi:10.3390/jpm12040571_

Round 1
Reviewer 1 Report
This review provides to the reader a general overview on the use of allogenic hematopoietic stem cell transplantation as a curative option for patients with primary myelofibrosis. Overall, this review has topical relevance and has the potential to be truly useful to the readers. Revision is needed to make it more cohesive and insightful.
Specific comments:
In the Introduction the authors should mention that current diagnosis of PMF is based on the 2016 WHO-criteria reported by Arber DA, et al. Blood. 2016;127(20):2391–405 and Barbui T, et al. Blood Cancer J. 2018;8:15.
The concept of considering allo-HSCT as a treatment for younger PMF patients, regardless of their risk score should be mitigated and discussed further.
The authors should report conditioning regimens including two alkylating agents with fludarabine, which have been shown to significantly reduce the risk of relapse when compared to regimen with one alkylating agent (either busulfan or thiotepa or melphalan) in combination with fludarabine. In addition, the value of splenic irradiation in the context of allo-HSCT should be mentioned and discussed.
The inclusion of a paragraph where the authors discuss the therapeutic strategies for PMF patients relapsing after an allogeneic transplant might be useful.
Page 2 lines 49-50: Please check the percentage of incidence of driver mutations in PMF patients (25% of MPL mutations in PMF?).
Please revise the data shown in Table 1. For example, in the DIPSS column: adverse points for Hbg <10 g/dL are = 2 instead than 1; DIPSS-plus column: “anemia requiring transfusion” should be “red cell transfusion need”.
Overall, the text should be carefully checked and revised (for example the content of sentences on lines 72-79, 101-102, and 225-228 is very unclear; on line 56 BCR-ABL should be BCR-ABL negative).
Author Response
- We added the WHO diagnostic criteria and cited Arber et al and Barbui et al in introduction section.
- Added comments in the patient factors and recommendations to highlight the consideration for transplantation in young patients regardless of risk score.
- Added data with 2 alkylating agents + fludarabine to conditioning section. As well as historic use of splenic irradiation in "disease related factors" section.
- Revised the percentages, ~ 5% with MPL. It was 20% with CALR per Rumi et al.
- Revised the data in table 1. Re-formatted the table for ease of viewing.
- Edits and revisions made for reader clarity.
Reviewer 2 Report
The manuscript entitled “The Use of Allogeneic Hematopoietic Stem Cell Transplantation in Primary Myelofibrosis” provides a comprehensive and informative review on the current state of the disease management for PMF. It appears convincingly interesting as to how the strategies which were earlier implemented only in higher risk patients including allo-HSCT, have been observed to have a better clinical outcomes in all patient categories in terms of survival and transformation to leukemic aberrations. The manuscript is suitable for publication. Just a few comments which are listed below:
-Line 35: “Laboratory abnormalities are a hallmark of this condition”. Kindly expand this narrative with specific biomarkers details.
-Please screen the manuscript for minor language issues. Some minor grammatical errors appear such as:
Line 39: Replace “They” with “these”
Line 71: Add “differ” after the outcomes
-A table that includes low and high-risk patient groups and summarizing the available treatment/management strategies with some statistical data (in favor or against) would have a greater impact on the subject of discussion.
Author Response
- Provided a few sentences about laboratory abnormalities and why they may vary from patient to patient.
- Grammatical errors were corrected.
- Added a table with low and high risk treatment options